# Non-Specific Pleuritis After Medical Thoracoscopy: The Portrait of an Open Issue and Practical Hints for Its Management

**DOI:** 10.3390/biomedicines13081934

**Published:** 2025-08-08

**Authors:** Matteo Daverio, Mariaenrica Tinè, Umberto Semenzato, Roberta Prevedello, Matteo Dalla Libera, Elisabetta Cocconcelli, Elisabetta Balestro, Marco Damin, Paolo Spagnolo, Davide Biondini

**Affiliations:** 1Department of Cardiac, Thoracic, Vascular Sciences and Public Health, University of Padova, 35128 Padova, Italy; matteo.daverio@unipd.it (M.D.); umberto.semenzato@aopd.veneto.it (U.S.); roberta.prevedello.1@studenti.unipd.it (R.P.); matteo.dallalibera@studenti.unipd.it (M.D.L.); elisabetta.cocconcelli@aopd.veneto.it (E.C.); elisabetta.balestro@aopd.veneto.it (E.B.); marco.damin@aopd.veneto.it (M.D.); paolo.spagnolo@unipd.it (P.S.); davide.biondini@unipd.it (D.B.); 2Department of Medicine, University of Padova, 35128 Padova, Italy

**Keywords:** lung cancer, mesothelioma, pleural effusion

## Abstract

**Background/Objectives:** Up to one third of pleural biopsies performed during medical thoracoscopy (MT) are labelled as non-specific pleuritis (NSP). The histological diagnosis of NSP has long been worrisome for pulmonologists, with the potential to evolve into a life-threatening condition. The aim of this study was to identify clinical and biological predictors for patients with a diagnosis of NSP to guide clinical decisions. **Methods**: Baseline, procedural and follow-up data of NSP patients were retrospectively analysed to identify potential outcome predictors. **Results**: Of the 272 patients who underwent MT, 192 (71%) were diagnosed with malignancies, 9 (3%) with benign diseases and 71 (26%) with NSP. At follow-up, 17% were diagnosed with malignant disease and 21% with a benign condition and 62% remained idiopathic. A thoracoscopist’s evaluation of the pleural appearance reported a PPV of 28% and an NPV of 91% to predict malignancy. Patients with a subsequent diagnosis of malignancy tended to have a higher volume of fluid drained than those with persistently idiopathic NSP [2.7 litres (L) vs. 1.6 L *p* = 0.06]. A lymphocytic pleural effusion was more common in the malignant and idiopathic groups (63% and 60%, respectively) than the benign group (16%; *p* = 0.06 and *p* = 0.01). The three groups had a similar rate of effusion recurrence. Overall survival was higher in patients with idiopathic pleural effusion than in those with malignant (*p* = 0.04) or benign disease (*p* = 0.008). **Conclusions**: NSP diagnosis hides a malignancy in one in five cases, underlying the importance of closely following up these patients. The volume of drained pleural fluid, cell count and thoracoscopist’s impression may guide clinicians in the challenging management of patients with NSP.

## 1. Introduction

Medical thoracoscopy (MT) is a well-established technique that allows the direct evaluation of the pleural cavity for both diagnostic and therapeutic purposes (e.g., the relief of dyspnoea, pleural biopsies, pleurodesis) [1]. Due to its safety profile and high diagnostic accuracy, MT is considered the best solution for investigating suspicious pleural malignancies [2], an increasing issue in clinical practice [3]. However, in up to 35% of cases, the diagnosis cannot be established and pathologists usually describe this condition as an “idiopathic pleuritis”, “non-specific pleuritis” (NSP) or a “pleuritis of indeterminate cause” [4,5,6,7,8,9,10,11].

Patients with NSP are usually clinically stable, have the absence of weight loss, are afebrile and have a modest quantity of pleural effusion (<50% of the hemithorax involved) [12]. NSP is a diagnosis of exclusion, determined once the main causes of unilateral pleural effusion have been ruled out with a pleural biopsy. This non-specific definition may reflect different causes, from a procedural error in biopsy technique or a truly benign condition to a precancerous disease. In previous retrospective studies, 3% to 15% of NSP patients were diagnosed with cancer at follow-up [5,7,8,9,10,11,13].

Recently, a European multicenter study collected 175 cases of NSP; notably, 6% of NSP developed a neoplastic disease after a mean time of one year. In this cohort, the CT finding of a pleural mass was the only feature that demonstrated a significant association to malignancy, but not asbestos exposure or sonographic pleural thickening [10].

To date, no clinical, radiological or histological marker of NSP evolution towards a malignant disease has been reliable. Moreover, there is no consensus on how long these patients should be followed up once NSP diagnosis is made.

With this hazy background, we aimed to better characterise patients with a histologically confirmed diagnosis of NSP in a high-volume pleural clinic, with a particular focus on potential predictors of malignant progression in order to improve NSP patients’ management.

## 2. Materials and Methods

### 2.1. Patients

This is a retrospective, single-centre study on 272 consecutive patients that underwent medical thoracoscopy (MT) for an undefined unilateral pleural effusion at the Azienda Ospedale-Università of Padova from January 2018 to March 2022. The data were consecutively recorded and collected from the reporting programme of the endoscopy suite (Eris Expiriva). For each patient, demographics, clinical data, results of pleural effusion analysis, thoracoscopic procedure and follow-up data in our pleural clinic were collected. Patients with histological diagnoses of NSP were included. Patients with active malignancies identified strictly before or after performing MT were excluded. Patients were classified according to the final diagnoses at follow-up in three groups: malignant (M), benign (B) or idiopathic (I).

### 2.2. Thoracoscopic Procedure

MT was performed by respiratory physicians experienced in the procedure (MD; DB). Briefly, patients were placed in a lateral decubitus position lying down on the healthy side; following ultrasound evaluation, conscious analgesia or deep sedation was administered. Local anaesthesia with lidocaine infusion was administered in the skin and in underlying structures in the rib space between the mid axillary line and the breast line. A 10 mm incision was made, a simple blunt dissection of chest muscles was performed, and a trocar was inserted into the pleural space. Pleural fluid was evacuated using a sterile suction catheter and samples were sent for cytological, microbiological and cell count analysis. The parietal, diaphragmatic and visceral pleura were macroscopically examined using a rigid thoracoscope (Karl Storz Endoscope, Tuttlingen, Germany) with video assistance and pleural biopsies were taken with overriding biopsy forceps through a rigid thoracoscope. When a malignant pleural effusion was assumed as the most probable diagnosis (according to the clinical data and macroscopical appearance), talc pleurodesis was performed under direct control (the poudrage technique) and a large bore chest tube was inserted (24–28 French).

### 2.3. Diagnostic Criteria for NSP

NSP was diagnosed if the histology showed follicular pleuritis, chronic pleuritis, non-specific pleuritis, and mesothelial hyperplasia in the absence of bacterial infection, pleural vasculitis, granulomas and neoplastic pleural infiltration. Patients with active malignancies identified strictly before or after performing MT were excluded.

### 2.4. Statistical Analysis

Continuous variables are described as the mean and standard deviations, whereas categorical variables were expressed as absolute numbers and percentage. The chi square test and Fisher *t* test (*n* < 5) for categorical variables and the Mann–Whitney U or Kruskal–Wallis test for continuous variables were used, as appropriate. Overall survival was calculated from medical thoracoscopy to death with data censored on 30 March 2023. The cumulative survival rate was calculated using the Kaplan–Meier method and the difference in the survival time between the groups was assessed with a log-rank test. *p* values below 0.05 were considered statistically significant.

## 3. Results

### 3.1. Clinical and Thoracoscopic Characteristics

Two-hundred seventy-two patients underwent MT between January 2018 and March 2022 at the Azienda Ospedale Università of Padova (Figure 1).

In 192 (71%) cases, pleural biopsies revealed a malignancy, in 9 (3%) cases, pleural biopsies revealed a benign disease (5 tuberculous pleuritis, 1 parapneumonic effusion, 1 empyema, 1 asbestos-related pleuritis and 1 amyloidosis), and in 71 (26%) cases, non-specific pleuritis (NSP) was reported. 

Among the NSP patients (*n* = 71), 13 (18%) were diagnosed with advanced cancer before performing thoracoscopy and were excluded to avoid the inclusion of false negative or paramalignant effusions. The study population therefore included 58 patients with NSP.

They were mainly men (72%) with a mean age of 75 years, almost half of them were current smokers and nearly one-third were exposed to occupational hazards (nine patients reported asbestos exposure) (Table 1).

After MT and the histological diagnosis of NSP, patients were followed up in our pleural clinic for a mean time of 22 months (minimum 40 days and maximum 5 years). During follow-up, 10 patients (17%) were diagnosed with malignant disease and 12 (21%) with benign disease and 36 (62%) remained idiopathic. Clinical characteristics were similar in the three groups (Table 1).

According to the final diagnosis, MT accuracy in idiopathic NSP diagnosis was 87.1% (95% CI, 82.6–90.9), sensitivity 100% (95% CI, 90.3–100), specificity 85.2% (95% CI, 79.9–89.5), PPV 50.7% (95% CI, 43.1–58.3), NPV 100% (95% CI, 98.2–100), the positive likelihood ratio 6.74 (95% CI, 4.9–9.2), and the negative likelihood ratio 0.

Among patients with malignancies, four received a diagnosis of malignant pleural mesothelioma (MPM) and six of other solid cancers [two non-small cell lung cancer not otherwise specified (NSCLC-NOS), one squamous cell carcinoma, one gastric cancer, one renal cell cancer and one ovarian cancer]. The mean time to cancer diagnosis was 6 ± 1 months.

Among benign cases, four were related to chronic heart failure, three to asbestos-related pleuritis, one to hepatic hydrothorax, one to non-tuberculous mycobacterial pleuritis, one to graft-versus-host disease (GVHD) in a patient following haematopoietic cell transplantation for myelofibrosis, one to systemic serositis and one to IgG4 disease (Table 2).

The intraprocedural variables are summarised in Table 3.

The main pleural fluid appearance was serous (64%) and haemorrhagic in the remaining cases (36%), and there were no differences in the pleural fluid appearance in the three subgroups. The prevalent macroscopic appearance of the pleura in malignant disease was nodular (55%), whereas in the idiopathic group, pleural thickening was the prevalent macroscopic appearance (32%) and nodules were seen in 12 cases (35%) (Figure 2).

However, these differences were not statistically relevant between the subgroups.

Thoracoscopic talc poudrage was performed in 25 patients (43%), especially in patients with NSP that turned out to be a malignant disease at follow-up (70%), compared to benign and idiopathic diseases (33% and 38%, *p* = 0.09 and *p* = 0.08, respectively) (Table 3). According to the data presented in Table 3, the thoracoscopist’s decision to perform talc poudrage (driven by pre-MT clinical/radiological data and confirmed intraprocedurally by macroscopic pleural appearance) had a sensitivity of 70.0% (95% CI, 34.8–93.3), a PPV of 28.0% (95% CI, 18.4–40.17), a specificity of 62.5% (95% CI, 47.4–76.1), and an NPV of 91% (95% CI, 79.1–96.4) to distinguish malignant from non-malignant causes.

### 3.2. Pleural Effusion Analysis 

Pleural fluid analysis was performed in 53/54 patients. The mean amount of pleural fluid drained tended to be higher in the malignant disease group than in the idiopathic group [2.7 litres (L) vs. 1.6 L *p* = 0.06] (Table 3). Pleural fluid analysis revealed the presence of an exudate in all malignant cases and in 89% of idiopathic NSP. This percentage was significantly lower in the benign disease group (58%; *p* = 0.04 and *p* = 0.05, respectively). There was a significantly higher number of patients with lymphocytic pleural effusion (the pleural lymphocyte rate > 0.5 of the total cell number) in the malignant and idiopathic groups (63% and 60%, respectively) compared to the benign group (16%; *p* = 0.06 and *p* = 0.01, respectively). Eosinophilic pleural effusion was detectable in only two patients, one in the malignant and one in the idiopathic disease group (Table 4).

### 3.3. Follow-Up Evaluation at Pleural Clinic

Only 37/58 (64%) patients of our population were followed up and, when appropriate, underwent therapeutic procedures in our pleural clinic. The remaining 21 were monitored in other centres who shared their follow-up and survival data with our centre. In total, 19/37 patients (50%) had at least one recurrence of pleural effusion within a median time of 197 (60–427) days. Among these, eight had chemical pleurodesis. There were no differences in the recurrence rate between the malignant, benign and idiopathic disease groups (50%, 33% and 60%, respectively). Recurrences were mainly treated with medical therapy (10/19; 53%). Three patients underwent surgical video-assisted thoracoscopy (VATS), two patients underwent thoracentesis, two patients underwent talc slurry pleurodesis and two did not require any treatment (Table 5).

### 3.4. Survival Analysis

The survival analysis in the overall population covered a median time of 12.3 (5.4–30.6) months. Regarding the three different groups, 78% of patients in the idiopathic group survived for at least one year after thoracoscopy, while in the other groups, this percentage tended to be lower (50% in patients with malignancies [*p* = 0.03] and 58% in those with benign conditions [*p* = 0.05]. Overall survival was significantly higher in patients with idiopathic pleural effusion than in those with malignant (HR 3.16 CI 1–10; *p* = 0.04) or benign disease (HR 2.7 CI 1–7; *p* = 0.008) (Figure 3).

## 4. Discussion

This study aimed to identify potential predictors of malignant evolution in patients with a diagnosis of non-specific pleuritis (NSP) at pleural biopsy. To this aim, baseline and follow-up clinical data, as well as procedural variables, were retrospectively collected, and the potential association with the subsequent development of a benign, malignant or persistently idiopathic condition was investigated.

One in five patients undergoing medical thoracoscopy in our centre had a diagnosis of NSP. The incidence of this entity, which often creates more questions than answers, has been observed to reach peaks of 35% in some cohorts [4,5,6,7,8,9,10,11], representing an enigma for pulmonologists who have neither clear guidelines on how to manage it nor reliable outcome predictors to tailor NSP patients’ follow-up.

In our cohort, 10/58 (17%) of patients had a subsequent diagnosis of pleural malignancy. Such a worrisome incidence of malignant evolution at follow-up is one of the highest reported [4,5,7,8,9,10,11,13] and could be related to a “selection bias”. Indeed, as a part of the thoracic oncology multidisciplinary team, oncologists and thoracic surgeons often refer patients with pleural effusions highly suspicious of being neoplastic to our pleural clinic [14]. Four of these patients (40%) developed malignant pleural mesothelioma, confirming the insidious nature of this neoplasm that could have a temporally and spatially heterogenous involvement of the pleural cavity [4,5,6,7,8,9,10,11,13,15].

Looking for potential predictors of subsequent outcomes in NSP, we analysed baseline clinical features of our patients. The three subgroups had an evenly distributed predominance of male smokers older than 70 years old. Of interest, almost all of them were former or current smokers, the highest rate ever reported in NSP patients [4,5,6,7,8,9,10,11,13]. Asbestos exposure, common in our cohort (20%), did not increase the risk of neoplastic evolution amongst the observed NSP patient, confirming its previously reported inaccuracy as a malignancy predictor in this subset of patients [4,5,10].

When we focused on procedural variables, we could observe that NSP patients who had a subsequent malignancy were characterised by a larger amount of pleural effusion drained during MT compared to the non-neoplastic ones. This novel evidence is difficult to explain, especially if we consider that the study population has the same histological pattern. We might hypothesise that in non-neoplastic NSP effusion, a more inflamed pleura may provoke earlier pain than in the neoplastic infiltration, leading to an earlier medical evaluation, even though chest pain is reported to be correlated to malignant involvement [4,7,11]. As an alternative, malignant NSP may have an intrinsic propensity to produce more fluid [13,16].

Conversely to this relation with pleural fluid volume and malignant NSP evolution, the suspicious macroscopic appearance has already been proposed in the risk stratification process of NSP patients, with inconsistent confidence [17]. Indeed, in contrast with Yang et al. and Yu et al. [7,8], our data go along with those from Davies et al. [5] suggesting that macroscopic findings alone, as pleural nodules, do not provide a reliable hint to neoplastic evolution.

On this line, we found that, when asked to report its impression on the aspect of the pleural space, the thoracoscopist’s accuracy on predicting malignancy was disappointing. Better performance was assessed when suspecting a non-malignant evolution, suggesting that the thoracoscopist’s impression is more reliable in benign/indeterminate diseases. In these cases, a “wait and see” approach could be preferable, while in the malignant suspicion, the management could be debatable. These data partially confirm Metintas et al.’s [4] findings, albeit with less confidence in the potential role of the physician in suspicions of malignancy. 

Another clue of a non-benign aetiology could be provided by the pleural fluid analysis. We systematically investigated pleural effusion cell count and, for the first time, found a significantly higher percentage of patients with pleural fluid lymphocytosis (lymphocytes >50%) in those with malignant evolution and idiopathic NSP compared to the benign evolution. This could be due to the predominance of exudates among those that had a malignant evolution (100%) and idiopathic NSP (89%), both sustained by underlying chronic inflammation, compared to the benign evolution of NSP (58%), and by the small size of the benign subgroup, which does not include typically lymphocytic effusions as seen with tuberculosis or rheumatoid arthritis [18].

In patients with neoplastic evolution, the mean time to cancer diagnosis was six months, shorter than other reports of almost one year [7,8,10,19]. These data underline that at least in this period after the NSP diagnosis, patients should be carefully followed up with multiple evaluations, especially to rule out neoplastic evolution [10,20]. Of note, we had two diagnoses of malignancy after the first year, suggesting that a one-year follow-up could still be not enough to exclude a neoplastic genesis of NSP. Unfortunately, our study could not answer the key questions of “how long” and “how often” these patients require check-ups, but it strongly supports the development of a dedicated therapeutic path, such as visiting a pleural clinic, to personalise patient monitoring.

Another important finding of this study was that one in five (21%) histologic NSP cases later revealed an underlying benign condition. Even though hidden malignancies are of the utmost concern when pleural biopsies diagnose NSP, we observed a significantly reduced survival rate, even amongst patients who received a delayed diagnosis of a benign condition compared to those who were confirmed to have idiopathic NSP at follow-up. This result reflects the severity of the underlying condition, the “benignity” of which does not reassure either the doctor or the patient but rather suggests the importance of a close follow-up. 

A final concern raised in the study is that the accuracy of MT, excellent in diagnosing malignant pleural effusion [2], drops to 87% for idiopathic NSP.

Overall, the histological diagnosis of NSP represents a challenging finding that needs a deep diagnostic investigation and meticulous follow-up. Pleural effusion recurrence was a common complication at follow-up, regardless of the subsequent diagnosis. What really differed amongst the three subgroups was the survival, significantly higher in idiopathic NSP compared to both malignant and benign disease (*p* = 0.04 and *p* = 0.008, respectively). Such evidence underlines the importance of pursuing a correct final diagnosis in these patients. Indeed, if the advantages of detecting a tumour early are obvious, the fact that benign conditions are characterised by poorer outcomes could be unexpected. Nonetheless, our observations, in line with other data [9], are due to the critically severe conditions identified—predominantly chronic heart failure—supporting the hypothesis that a pleural effusion of non-malignant aetiology may reflect the end stage of benign diseases [21].

We acknowledge that our study is limited by its retrospective and monocentric design. Moreover, the sample size is limited and the subsets analysed are somewhat “arbitrary”, being defined according to clinical expertise and potentially incorrect for those who died earlier. On the other hand, to the best of our knowledge, this is the first study that has investigated the quality and quantity of pleural fluid in patients with NSP and it provides detailed follow-up data on mortality and the rate of recurrence. Prospective data are needed to evaluate other diagnostic biomarkers and to determine the timing and modality of follow-up.

## 5. Conclusions

Non-specific pleuritis diagnosis hides a malignancy in one in five cases, and even when it masks a benign condition, it carries a significantly worse prognosis than an idiopathic condition. These findings underscore the importance of a careful follow-up of these patients, ideally in a dedicated setting and with more frequent check-ups in the first six months. The volume of drained pleural fluid, cell count and thoracoscopist’s impression may guide clinicians in the challenging management of patients with NSP.

## Figures and Tables

**Figure 1 biomedicines-13-01934-f001:**
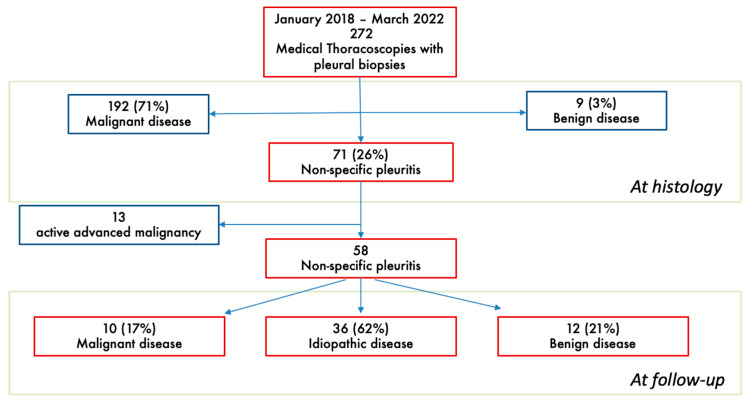
Flowchart of study.

**Figure 2 biomedicines-13-01934-f002:**
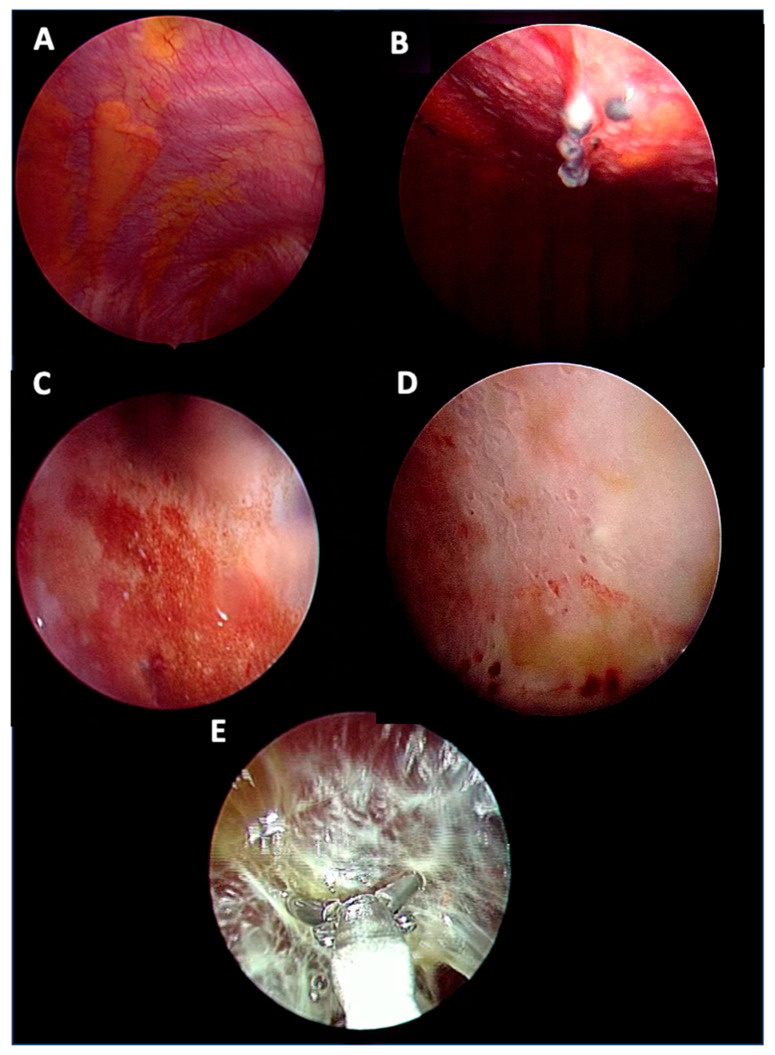
Thoracoscopic appearance: (**A**) normal; (**B**) nodular; (**C**) hyperemia; (**D**) thickening; (**E**) septations.

**Figure 3 biomedicines-13-01934-f003:**
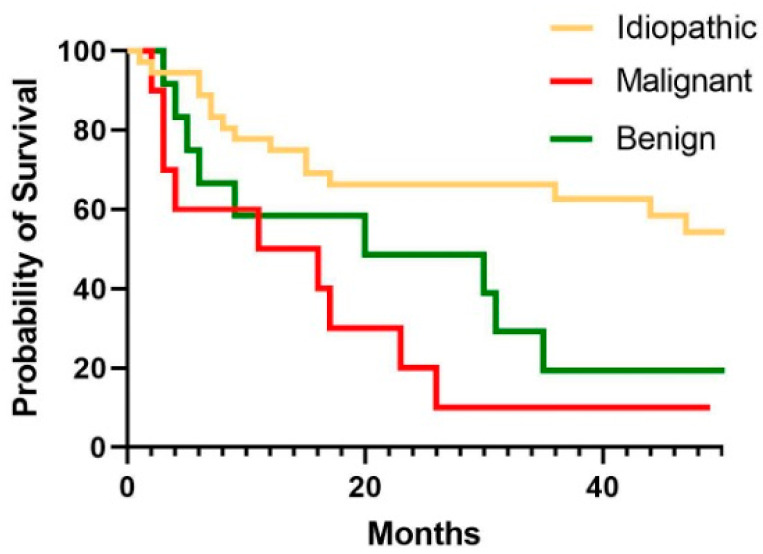
Kaplan–Meyer curves showing the survival of the three subgroups: malignant disease, benign disease and idiopathic disease.

**Table 1 biomedicines-13-01934-t001:** Clinical characteristics of the overall population and of the three subgroups of malignant disease, benign disease and idiopathic disease. Values are expressed as numbers and (%) or mean and standard deviation. To compare the demographic data between malignant, benign and idiopathic disease, the Chi square test and Fisher *t* test (*n* < 5) for categorical variables and the Mann–Whitney U test for continuous variables were used, as appropriate.

	Overall Population 58	Malignant Disease 10	Benign Disease 12	Idiopathic Disease 36	*p* Value
**Male, N (%)**	42 (72)	7 (70)	9 (75)	26 (72)	n.s.
**Age, years**	75 ± 10	75 ± 6	74 ± 9	74 ± 11	n.s.
**Smoking history, N (%)** Former Current Never	21 (44) 25 (52) 2 (4)	5 (50) 5 (50) 0	3 (37) 5 (63) 0	13 (43) 15 (50) 2 (6)	n.s.
**Pack years**	22 ± 17	20 ± 9	22 ± 10	22 ± 22	n.s.
**Professional exposure, N (%)** Asbestos	18 (40) 9 (20)	4 (44) 0	1 (20) 1 (20)	4 (13) 8 (26)	n.s.

**Table 2 biomedicines-13-01934-t002:** Different aetiologies of the benign disease group. 1NTM: nontuberculous mycobacteria; 2 GVHD: graft-versus-host disease.

Benign Effusion Aetiologies	N° (%)
Chronic heart failure	4 (33%)
Asbestos-related pleuritis	3 (25%)
Hepatic hydrothorax	1 (8%)
NTM^1^ pleuritis	1 (8%)
GVHD^2^ following stem cell transplantation	1 (8%)
Serositis	1 (8%)
Immunoglobulin G4-related disease	1 (8%)

**Table 3 biomedicines-13-01934-t003:** Intraprocedural variables in the overall population and of the three subgroups of malignant disease, benign disease and idiopathic disease. Values are expressed as numbers and (%) or mean and standard deviation. To compare the demographic data between malignant, benign and idiopathic disease, the Chi square test and Fisher *t* test (*n* < 5) for categorical variables and the Mann–Whitney U test for continuous variables were used. ^ The data refer to 54 patients in which we had a description of macroscopic pleural appearance. * Comparison between malignant and idiopathic (*p* = 0.06). ** Comparison between malignant and benign (*p* = 0.09) and idiopathic (*p* = 0.08).

	Overall Population 58	Malignant Disease 10	Benign Disease 12	Idiopathic Disease 36
**Fluid appearance** Serous, N (%) Haemorrhagic, N (%)	37 (64) 21 (36)	7 (70) 3 (30)	7 (58) 5 (42)	23 (64) 13 (36)
**Thoracoscopic appearance ^, N (%)** Nodular Hyperaemia Septations Thickening Normal	54 19 (35) 7 (13) 7 (13) 13 (24) 8 (15)	9 5 (55) 2 (23) 1 (11) 0 (0) 1 (11)	11 2 (18) 2 (18) 2 (18) 2 (18) 3 (28)	34 12 (35) 3 (9) 4 (12) 11 (32) 4 (12)
**Volume of drained effusion (L)**	1.8 ± 1.1	2.7 ± 1.6 *	1.8 ± 1.1	1.6 ± 0.9
**Talc poudrage** **N (%)**	25 (43)	7 (70) **	4 (33)	14 (38)

**Table 4 biomedicines-13-01934-t004:** A pleural fluid analysis of the overall population and of the three subgroups of malignant disease, benign disease and idiopathic disease. Values are expressed as numbers and (%). To compare the demographic data between malignant, benign and idiopathic disease, the Chi square test and Fisher *t* test (*n* < 5) for categorical variables and the Mann–Whitney U test for continuous variables were used. * Significantly different from malignant and idiopathic, respectively, *p* = 0.05 and *p* = 0.04. *p* values refer to Fisher’s exact test. ** Significantly different from malignant and idiopathic, respectively, *p* = 0.06 and *p* = 0.01. *p* values refer to Fisher’s exact test.

	Overall Population 53	Malignant Disease 8	Benign Disease 12	Idiopathic Disease 33
**Pleural Fluid** Exudate (%) Lymphocytes > 50% (%) Eosinophils > 10% (%)	44 (83) 27 (51) 2 (4)	8 (100) 5 (63) 1 (13)	7 (58) * 2 (16) ** 0 (0)	29 (89) 20 (60) 1 (3)

**Table 5 biomedicines-13-01934-t005:** A follow-up evaluation of the overall population and of the three subgroups of malignant disease, benign disease and idiopathic disease. FU: follow-up; MT: medical thoracoscopy; VATS: video-assisted thoracoscopic surgery. ^ The data refer to 37 patients with complete follow-up data; * significantly different from malignant and benign. *p* = 0.03 and 16%; *p* = 0.05, respectively. *p* values refer to Fisher’s exact test; ** tended to be higher than malignant and benign (*p* = 0.08).

	Overall Population 37	Malignant Disease 8	Benign Disease 9	Idiopathic Disease 20
**Recurrency, N (%)** **Time from MT to recurrency ^ (days)** **Recurrency treatment ^** **Medical therapy** **Thoracentesis** **Chest drain** **VATS** **No therapy**	19 (50) 197 (0–427) 10 (59) 2 (11) 2 (11) 3 (17) 2 (11)	4 (50) 248 (67–429) 2 (50) 0 1 (25) 1 (25) 0	3 (33) 581 (393–769) 2(66) 0 1(33) 0 0	12 (60) 78 (17–214) 6 (60) 2 (20) 0 2 (20) 2 (20)
**1-year survival, N (%)**	40 (69)	5 (50)	7 (58)	28 (78) **
**Survived at FU, N (%)**	21 (36)	1 (10)	2 (16)	18 (50) *
**Time from MT to death (days)**	371 (163–919)	315 (83–591)	434 (149–963)	405 (195–1353)

## Data Availability

The dataset analysed during the current study is available from the corresponding author upon reasonable request.

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
