# Peer review of "Non-Specific Pleuritis After Medical Thoracoscopy: The Portrait of an Open Issue and Practical Hints for Its Management"

_biomedicines, 2025, doi:10.3390/biomedicines13081934_

Round 1
Reviewer 1 Report
Comments and Suggestions for Authors
Dear Editor,
Thank you for the opportunity to read and evaluate the manuscript titled "Non-specific pleuritis after medical thoracoscopy: portrait of an open issue and practical hints for its management." My comments regarding the manuscript are as follows:
Non-specific pleuritis is a current and clinically relevant topic in respiratory medicine, warranting further investigation. However, I believe several important revisions are needed in the manuscript.
- Abstract:In the Results section, I suggest beginning with the total number of patients who underwent medical thoracoscopy, followed by a breakdown of diagnostic outcomes. Considering the designation of "non-specific pleuritis" (NSP), the presentation of the outcome distribution should be revised, as discussed in more detail below.
- Introduction:This section contains excessive background information, which seems unnecessary for the scope of this manuscript. Specifically, the opening sentences of the first paragraph could be reconsidered. The second paragraph could be rewritten more concisely to focus the reader's attention on the issue at hand.
- Results:The presentation of findings is not sufficiently clear. Both the flow chart and the corresponding text require revision:
- The current flow chart and the diagnostic distribution in the tables are somewhat confusing for the reader.
- The flow chart should begin with the total number of patients (n=272) who underwent medical thoracoscopy and then categorize them into three diagnostic groups with respective counts and percentages: (a) malignant pleural diseases, (b) granulomatous diseases (e.g., tuberculous pleurisy), and (c) non-specific pleuritis (NSP).
- Among NSP cases, false negatives should be reported with their number and specific final diagnoses. The remaining true negatives—those definitively diagnosed with benign NSP or classified as idiopathic—should also be reported, along with their counts and proportions. Within the true negative group, benign diseases should be subclassified, and the proportion of idiopathic NSP should be explicitly presented, as it is clinically significant.
- NSP is a histopathological diagnosis made in the absence of malignancy or granulomatous inflammation. Thus, all benign and idiopathic cases are encompassed within this category. It is crucial to distinguish false negatives from idiopathic NSP to enable meaningful discussion. The findings of this study are valuable, but the clarity and structure of the results could be enhanced to reflect their importance better.
- Once this classification is established, the sensitivity, specificity, positive and negative predictive values, likelihood ratios (-LR and +LR), and diagnostic accuracy of medical thoracoscopy can be calculated and presented accordingly.
- Follow-up Duration:The follow-up duration for NSP patients should be clearly defined, including both the minimum and maximum follow-up periods. This feature is especially relevant for cases that were later diagnosed with mesothelioma.
- Table 3:The term "bloody pleural fluid" should be clarified—does it refer to hemorrhagic pleural effusion?
- Also, in Table 3,The indication for talc pleurodesis should be explained, particularly in patients with benign or idiopathic NSP. In idiopathic NSP, pleural effusions are generally expected to resolve spontaneously.
- Figure 2:The thoracoscopic images, particularly panel (b), "nodular pleural appearance," and panel (d), "pleural thickening," do not appear to represent the described findings accurately. No clear nodularity is visible, and the areas labeled as pleural thickening appear to reflect pleural adipose tissue rather than pathological thickening.
Reviewer 2 Report
Comments and Suggestions for Authors
Dear Authors;
Below, I summarize my key recommendations for improving the paper’s clarity, rigor, and impact. The study highlights important gaps in NSP management, particularly the risk of malignant progression.
Some terms like "idiopathic pleuritis" and "non-specific pleuritis" should be define clearly. You should state state how this study advances prior work in the introduction part.
The intervals for monitoring is not clear. Moreover, you may address selection bias and small subgroup sizes as well as a follow-up algorithm for high-risk NSP. A clear recommendation for readers is missing at the end of this study.
Sincerely,
